

# The corepressor NCOR1 and OCT4 facilitate early reprogramming by suppressing fibroblast gene expression

Georgina Peñalosa-Ruiz, Klaas W. Mulder and Gert Jan C. Veenstra

Department of Molecular Developmental Biology, Faculty of Science, Radboud Institute for Molecular Life Sciences, Radboud University, Nijmegen, The Netherlands

Corresponding authors
Klaas W. Mulder,
k.mulder@science.ru.nl
Gert Jan C. Veenstra,
g.veenstra@science.ru.nl

## ABSTRACT

Reprogramming somatic cells to induced pluripotent stem cells (iPSC) succeeds only in a small fraction of cells within the population. Reprogramming occurs in distinctive stages, each facing its own bottlenecks. It initiates with overexpression of transcription factors OCT4, SOX2, KLF4 and c-MYC (OSKM) in somatic cells such as mouse embryonic fibroblasts (MEFs). OSKM bind chromatin, silencing the somatic identity and starting the stepwise reactivation of the pluripotency programme. However, inefficient suppression of the somatic lineage leads to unwanted epigenetic memory from the tissue of origin, even in successfully generated iPSCs. Thus, it is essential to shed more light on chromatin regulators and processes involved in dissolving the somatic identity. Recent work characterised the role of transcriptional corepressors NCOR1 and NCOR2 (also known as NCoR and SMRT), showing that they cooperate with c-MYC to silence pluripotency genes during late reprogramming stages. NCOR1/NCOR2 were also proposed to be involved in silencing fibroblast identity, however it is unclear how this happens. Here, we shed light on the role of NCOR1 in early reprogramming. We show that siRNA-mediated ablation of NCOR1 and OCT4 results in very similar phenotypes, including transcriptomic changes and highly correlated high-content colony phenotypes. Both NCOR1 and OCT4 bind to promoters co-occupied by c-MYC in MEFs. During early reprogramming, downregulation of one group of somatic MEF-expressed genes requires both NCOR1 and OCT4, whereas another group of MEF-expressed genes is downregulated by NCOR1 but not OCT4. Our data suggest that NCOR1, assisted by OCT4 and c-MYC, facilitates transcriptional repression of genes with high expression in MEFs, which is necessary to bypass an early reprogramming block; this way, NCOR1 facilitates early reprogramming progression.

## INTRODUCTION

Induced pluripotent stem cells (iPSCs) are generated in vitro by overexpressing factors OCT4, SOX2, KLF4 and c-MYC (OSKM factors) in somatic cells (*Takahashi & Yamanaka, 2006*). iPSCs have been successfully used in disease modelling and cell

transplantation, showcasing their relevance in regenerative medicine (*Inoue et al., 2014*). Despite significant improvements to the original protocol (*Di Stefano et al., 2016*; *Esteban et al., 2010*; *Vidal et al., 2014*), in general, only a small percentage of somatic cells become pluripotent (*Stadtfeld & Hochedlinger, 2010*). The reason is that, as embryonic development proceeds, cells gradually commit to a certain lineage and acquire full stability upon differentiation (*Perino & Veenstra, 2016*). Thus, in vivo, differentiated cells are not able to become other cell types and it is possible that such stability is, at least in part, conferred by chromatin state. Chromatin state defines the properties of regulatory elements as a function of DNA sequence, histone modifications, DNA methylation, chromatin architecture and the activity of transcription factor networks (*Perino & Veenstra, 2016*). To understand reprogramming and develop new or improve existent iPSC generation protocols, it is crucial to understand the interplay of chromatin modulators and transcriptional regulation.

Pluripotency acquisition involves silencing of somatic genes (*Polo et al., 2012*), and reactivation of the pluripotency network, which occurs in a stepwise manner (*Buganim et al., 2012*). The OSKM reprogramming transcription factors play different roles, but assist the two processes (*Chronis et al., 2017*). For instance, it has been shown that KLF4 and SOX2 bind to closed chromatin in MEFs (*Li et al., 2017*; *Soufi et al., 2015*). These binding events are related to opening up of silenced pluripotency enhancers (*Li et al., 2017*). It has also been suggested that opening up of pluripotency enhancers requires cooperative binding of at least two of the reprogramming factors (*Chronis et al., 2017*). There is evidence indicating that OCT4 binds to open (*Knaupp et al., 2017*; *Soufi et al., 2015*) as well as to closed chromatin in somatic cells (*Chronis et al., 2017*; *Donaghey et al., 2018*). However, the dynamics of these events are unknown and conclusions vary depending of the reprogramming conditions. OSK also are involved in transcriptional silencing of somatic genes, facilitating the relocation of somatic-specific transcription factors to loci co-occupied by OSK (*Chronis et al., 2017*). Thus, reprogramming factors facilitate the repression of the somatic programme but also the activation of the pluripotency network. This transcriptional duality is elicited by differential interaction with transcriptional coactivators or corepressors (*Mottis, Mouchiroud & Auwerx, 2013*).

The transcriptional corepressors NCOR1 (nuclear receptor corepressor 1) and its paralogue NCOR2 (silencing mediator of retinoic acid and thyroid hormone receptor or NCOR2), interact with transcription factors to suppress their target genes (*Mottis, Mouchiroud & Auwerx, 2013*). NCOR1/NCOR2 are essential non-enzymatic co-factors of histone deacetylases, especially HDAC3 (*You et al., 2013*) and mediate transcriptional repression of their targets via deacetylation of Lysine residues of histone tails (*Alland et al., 1997*; *Heinzel et al., 1997*). NCOR1/NCOR2 are essential for organism homeostasis and have non-redundant roles in metazoans, as mutation of each gene individually leads to embryonic lethality (*Mottis, Mouchiroud & Auwerx, 2013*). In fact, NCOR1/NCOR2 regulate several essential metabolic and cellular processes, such as circadian rhythm (*Alenghat et al., 2008*) and mitochondrial function (*Fan & Evans, 2015*). Recently, it was shown that NCOR1/NCOR2 interact with all four OSKM factors during

reprogramming from MEFs to iPSC (*Zhuang et al., 2018*). NCOR1/NCOR2 interaction with c-MYC promotes the repression of pluripotency genes in later reprogramming stages (*Zhuang et al., 2018*), imposing a barrier for iPSC generation. In addition, NCOR1/NCOR2 knockdowns induced transcriptional upregulation of somatic genes in the earliest reprogramming stages (*Zhuang et al., 2018*), arguing in favour of a dual role for these corepressors. However, it is unknown how these early effects of NCOR1/NCOR2 are brought about.

Previously, we performed a high-content imaging siRNA screening, combined with a secondary RNA-sequencing screen, to identify chromatin-associated regulators during early reprogramming from MEFs to iPSC (*Peñalosa-Ruiz et al., 2019*). Using this data from two orthogonal screens, we identified strong phenotypic similarities between NCOR1 and OCT4 knockdowns. We have identified functional interactions from knockdowns showing similar phenotypes before (*Mulder et al., 2012*; *Peñalosa-Ruiz et al., 2019*). Here we characterise the relationship of NCOR1 and OCT4 in early reprogramming. For such purpose, we first looked closer into the phenotypic similarities of both knockdowns, based on the high-content imaging. Then we set out to investigate the effect on the reprogramming transcriptome in *Ncor1* and *Oct4* knockdowns with RNA-sequencing. Finally, we compared our RNA-seq findings to published ChIP-seq and RNA-seq datasets. These analyses not only document the cooperation of OCT4 and NCOR1 in downregulating one set of somatic genes, they also show an antagonistic role in the regulation of another subset of somatic genes during early reprogramming.

## MATERIALS AND METHODS

### Data availability

New RNA-sequencing data generated for this article can be accessed via NCBI GEO database with the accession number GSE139376. High-content screening data can be downloaded as supplemental data from our previous study (*Peñalosa-Ruiz et al., 2019*). Other RNA-sequencing data generated in our lab previously, include the reprogramming siRNA non-targeting controls and time-course RNA-sequencing in reprogramming (*Peñalosa-Ruiz et al., 2019*). These can be accessed under a superseries with accession number GSE118680.

### MEF-to-iPSC reprogramming

Primary mouse embryonic fibroblasts (MEFs) were derived from mouse wild type strain C57/BL6 (Erasmus MC iPSC facility). Passage 0–1 MEFs were seeded at a density of 10,000 cells per cm$^2$ in MEF medium consisting of 15% FBS, 100 nM β-mercaptoethanol and 1% non-essential aminoacids (Thermo Scientific, Waltham, MA, USA). Next day, MEFs were transduced with the doxycycline-inducible mouse tet-STEMCCA (*Sommer et al., 2009*) and rtTA lentiviruses (Addgene # 20342) at an MOI of 1. One day after transduction, cells were either transfected with siRNAs or induced for reprogramming with reprogramming medium. This consisted of DMEM-high glucose (Thermo Scientific, Waltham, MA, USA) supplemented with 10% stem cell

grade FBS (Hyclone, Logan, UT, USA), 200 nM β-mercaptoethanol (Sigma, Kawasaki, Kanagawa, Japan), 1% sodium pyruvate (Thermo Scientific, Waltham, MA, USA), 2 μg·mL$^{-1}$ doxycycline, 3 μM Chiron (GSK3-inhibitor), 0.25 μM Alk5i (TGF-ß inhibitor), 50 μg·mL$^{-1}$ ascorbic acid and $1 \times 10^3$ U·mL$^{-1}$ LIF. Next day after adding reprogramming medium was considered reprogramming Day 1.

## siRNA transfections

Transfections were performed as described previously (*Peñalosa-Ruiz et al., 2019*). Briefly, before adding the cell suspension containing OSKM-transduced MEFs, multi wells were prepared with transfection mix. This consisted of a 40 nM pool of 3 different siRNAs per target mRNA, diluted in Optimem (Thermo Scientific, Waltham, MA, USA) together with RNAiMAX lipofectamine, according to the manufacturers instructions. After this incubation, cell suspensions were added to each well. A total of 24 h after transfections, reprogramming started by adding medium with doxycycline.

## High-content screening

High-content screening was performed as described before (*Peñalosa-Ruiz et al., 2019*). Briefly, transfected cells were cultured in Cell Carrier 96-well black plates (Perkin Elmer, Waltham, MA, USA), after 6 days, samples were fixed with 4% PFA and stained for CDH1 (Cell Signaling 14472) and SALL4 (29112; Abcam, Cambridge, UK) with DAPI counterstain. Plates were imaged in an Opera High-Content-Screening System (Perkin Elmer, Waltham, MA, USA). Colony segmentation was done in multiple $Z$-planes using SALL4 staining. Columbus Software (Perkin Elmer, Waltham, MA, USA) was used to extract all features in an automated fashion. $Z$-score normalisation per plate was applied. The normalised data for high-content features were used for knockdown-to-knockdown Pearson-correlation analysis.

## RNA isolation and RT-qPCR

RNA was isolated with RNA Micro-prep kit (Zymo Research, Irvine, CA, USA). Integrity was verified with Bioanalyzer and concentrations were determined with Nanodrop. For reverse transcription we used SuperScript III Kit (Thermo Scientific, Waltham, MA, USA) starting with 120–180 ng total RNA. For the RT-qPCR reaction 1–2 ng of cDNA were used in 20 μL reaction mix with Sybr Green mix ready to use (iQ-SYBR-Green Supermix; Biorad, Hercules, CA, USA). Relative gene expression was calculated with the ΔΔCt method using GAPDH as reference.

## RNA-sequencing library preparation and data analysis

Kapa-RNA HyperPrep kit with RiboErase (Roche; Kapa Biosystems, Wilmington, MA, USA) was used for ribosomal RNA depletion and library preparation, with 200 ng total RNA. Library amplification was performed for 10 cycles, according to manufacturer's instructions. Correct size distribution of 300 bp was verified with Bioanalyzer (Agilent Technologies, Santa Clara, CA, USA). Quality controls were performed for selected transcripts in each sample before and after library preparation by qPCR.

NextSeq500 Illumina platform was used for paired-end library sequencing, with 43 bp read length. STAR v.2.5.b (*Dobin et al., 2013*) was used to align reads to mouse genome assembly mm10. Data was normalised to log2-cpm with R package edgeR v.3.20.9 (*Robinson, McCarthy & Smyth, 2010*).

For differential gene expression analysis we used R Package DESeq2 v.1.18.1 (*Love, Huber & Anders, 2014*). Gene lists were filtered with a cut-off of $p$-adjusted value < 0.05.

Gene Ontology classification with differential gene lists was performed with web-based DAVID (*Huang, Sherman & Lempicki, 2009*), considering only categories with gene counts > 10 and $p$-adjusted value < 0.05.

### ChIP-seq and RNA-seq data integration

For ChIP-seq data integration, FASTQ files of ChIP-seq data were downloaded from NCBI GEO (GSE70736, GSE90893, GSE90895) and mapped to the mm10 genome assembly using BWA allowing one mismatch per read. NCOR1/NCOR2 peaks (*Zhuang et al., 2018*) were intersected with *siNcor1* day 6-upregulated genes (linux join, using gene symbols) to obtain genomic locations of interest. Heatmaps of ChIP-seq data were generated with fluff (*Georgiou & Van Heeringen, 2016*) with hierarchical clustering and RPKM normalisation. RNA-seq data (*Chronis et al., 2017*) was intersected with differentially expressed genes in our study (linux join, using gene symbols). Corresponding RNA-seq heatmaps were generated using seaborn clustermap with row clustering and row $z$-score normalisation. Venn diagrams were obtained using venn3 from the matplotlib_venn library. Hypergeometric probabilities were calculated using scipy.stats.

## RESULTS

### High-content screening reveals early reprogramming disruption upon *Ncor1* depletion

In a previous study, we performed a high-content siRNA screen, targeting 300 chromatin-associated factors during early reprogramming (Fig. 1A) (*Peñalosa-Ruiz et al., 2019*). We hypothesised that the onset of reprogramming would be associated with colony-level phenotypic changes and that these changes are affected upon gene perturbation. Moreover, using computational analysis of high-content imaging data, perturbed genes can be grouped according to their phenotypic similarities in a multidimensional space, rather than focusing on a single phenotype (*Boutros, Brás & Huber, 2006*). We used early pluripotency markers CDH1 and SALL4 to detect colonies at day 6 of reprogramming. Additionally, multiple colony features were extracted, including pluripotency marker expression levels, morphological traits such as colony roundness, area and symmetry, and many other colony texture and morphological features (*Peñalosa-Ruiz et al., 2019*). We selected 30 hits that were subjected to a secondary RNA-seq-based screening (*Peñalosa-Ruiz et al., 2019*) (overview in Fig. 1A). We have previously identified functional interactions from knockdowns showing similar phenotypes (*Mulder et al., 2012*; *Peñalosa-Ruiz et al., 2019*; *Tanis et al., 2018*). Using this rationale, we compared knockdown-to-knockdown correlations based on high-content phenotypes and also based on transcriptomes (Fig. 1B). We filtered for pairwise correlations showing

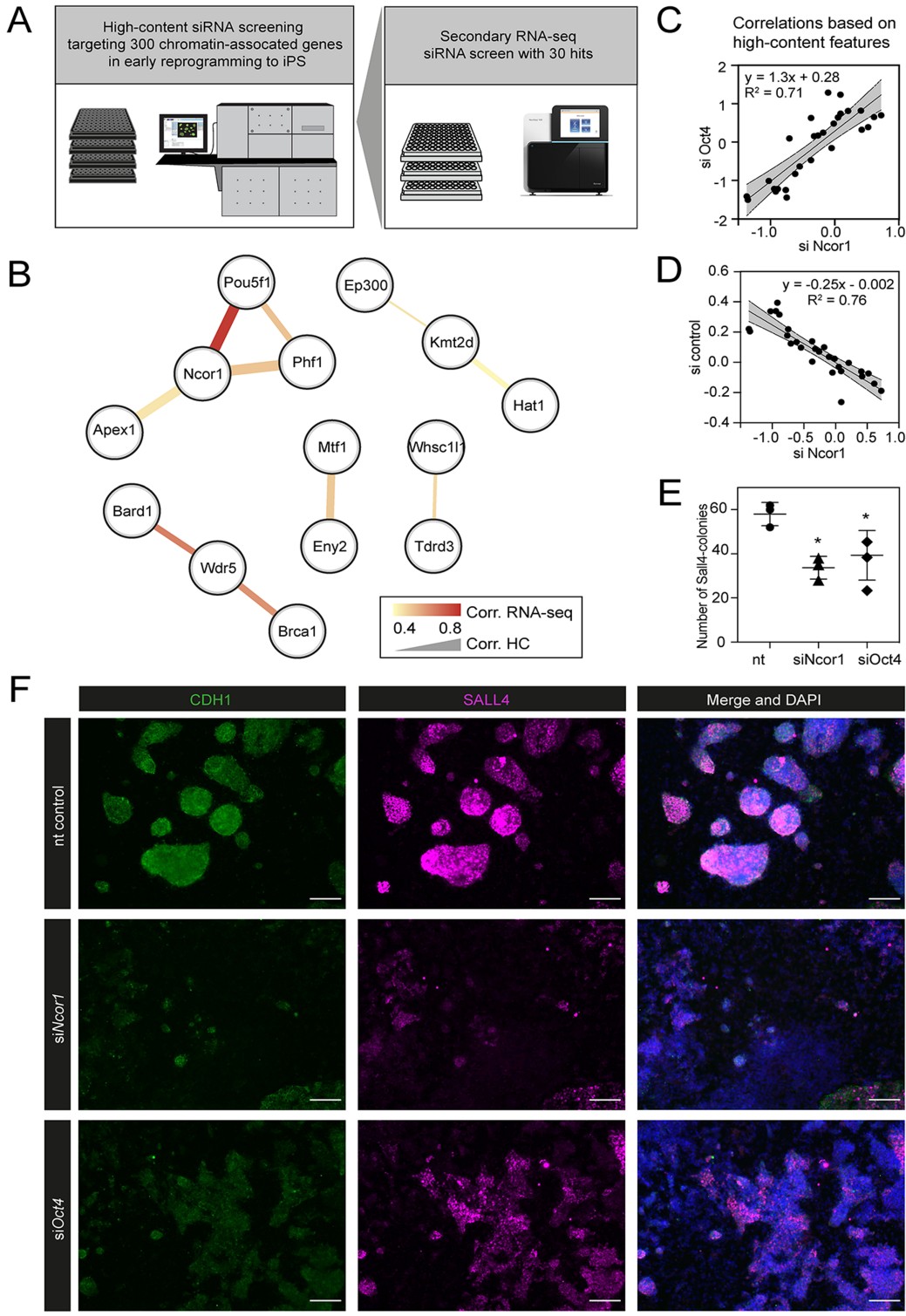

**Figure 1 High content screening identifies *Ncor1* as an early reprogramming modulator.** (A) Overview of experimental setup for high-content siRNA screening in early reprogramming, followed by secondary RNA-seq-based screening, from which *Ncor1* was selected as candidate to follow-up. (B) Gene network depicting correlations of high content colony phenotypes and RNA-seq profiles of genes targeted by siRNA. The nodes represent genes and the edges are pairwise Pearson correlations. The width of

**Figure 1 (continued)**
the edge represents correlation score by high-content imaging features and the colour of the edge
represents the correlation strength based on RNA-sequencing. (C and D) Scatterplots showing Pearson
correlations according to 27 selected high-content features between *Ncor1* and *Oct4* knockdowns (C) and
between *Ncor1* and non-targeting (nt) control (D). The middle line represents the best fit to linear
regression model and the grey shades the 95% confidence intervals (CI). The equation is the linear
regression model and $r^2$ is the correlation coefficient. (E) Reprogramming efficiencies (SALL4-positive
number of colonies) measured by in-cell western, compare *Ncor1* and *Oct4* knockdowns vs nt control.
*Ncor1* depletion impairs colony formation during early stages of reprogramming. Each data point
represents one independent transfection and lines represent mean ± SD. Statistical significance analysed
by One-way ANOVA, (*) = $p \leq 0.03$. (F) Representative high-content images showing that, at day 6,
*Ncor1* and *Oct4* depleted colonies display similar phenotypes, as compared to the non-targeting control.
Scale bar represents 150 µM. In magenta nuclear transcription factor SALL4 and in green E-Cadherin
(CDH1), counterstained with DAPI.               

highest scores in both the high-content and the RNA-seq screen (both $R > 0.4$).
We visualised the resulting potential interactions between genes (nodes) as networks,
considering both high-content imaging correlations (edge width) and transcriptome-based
correlations (edge colour) of all siRNA knockdowns. We previously uncovered a
functional interaction between BRCA1, BARD1 and WDR5 (*Peñalosa-Ruiz et al., 2019*).
The strongest correlation however is between NCOR1-OCT4, raising the possibility of a
functional interaction between these factors (Fig. 1B).

We verified that, based on high-content features, *siNcor1* positively correlated with
*siOct4*, but anti-correlated with the negative (nt, non-targeting) control (Figs. 1C and 1D).
Colony number analysis derived from the screen showed a significant reduction of
colony formation upon *Ncor1* depletion (Fig. 1E; Fig. S1). High-content images from
*Ncor1* and *Oct4* knockdowns showed very low CDH1 and SALL4 intensities and
compromised colony formation, when compared to the nt control. We validated this
finding using an independent In-Cell Western assay and confirmed that *siNcor1* and si*Oct4*
show similarly impaired reprogramming phenotypes (Fig. 1F; Fig. S1).

From these experiments we concluded that siRNA targeting of *Ncor1* results in an early
reprogramming phenotype that is similar to that caused by *Oct4* knockdown.

### *Ncor1* depletion affects transcriptional regulation of fibroblast identity

To gain insight in the molecular events that explain early reprogramming disruption by
*Ncor1* depletion, we performed gene expression analyses. We first asked whether *Ncor1*
mRNA expression followed a dynamic pattern during early reprogramming. For this,
*Ncor1* mRNA expression was quantified by RT-qPCR at different reprogramming
timepoints, from MEFs to reprogramming day 6. We found that *Ncor1* mRNA expression
was relatively stable throughout reprogramming, with slight variations (Fig. 2A). Then, we
sought to understand the role of NCOR1 in transcriptional dynamics during early
phases of iPSC reprogramming. For this purpose, we performed RNA-sequencing at day 3
and day 6 after expression of OSKM factors in MEFs transfected with control or *Ncor1*
targeting siRNAs, respectively. Differential gene expression analysis revealed 405
deregulated genes at day 6 (nt control vs *siNcor1*, adjusted *p*-value < 0.05, Figs. 2B and 2C).
The effect size was relatively small for most transcripts, suggesting that the strong defects

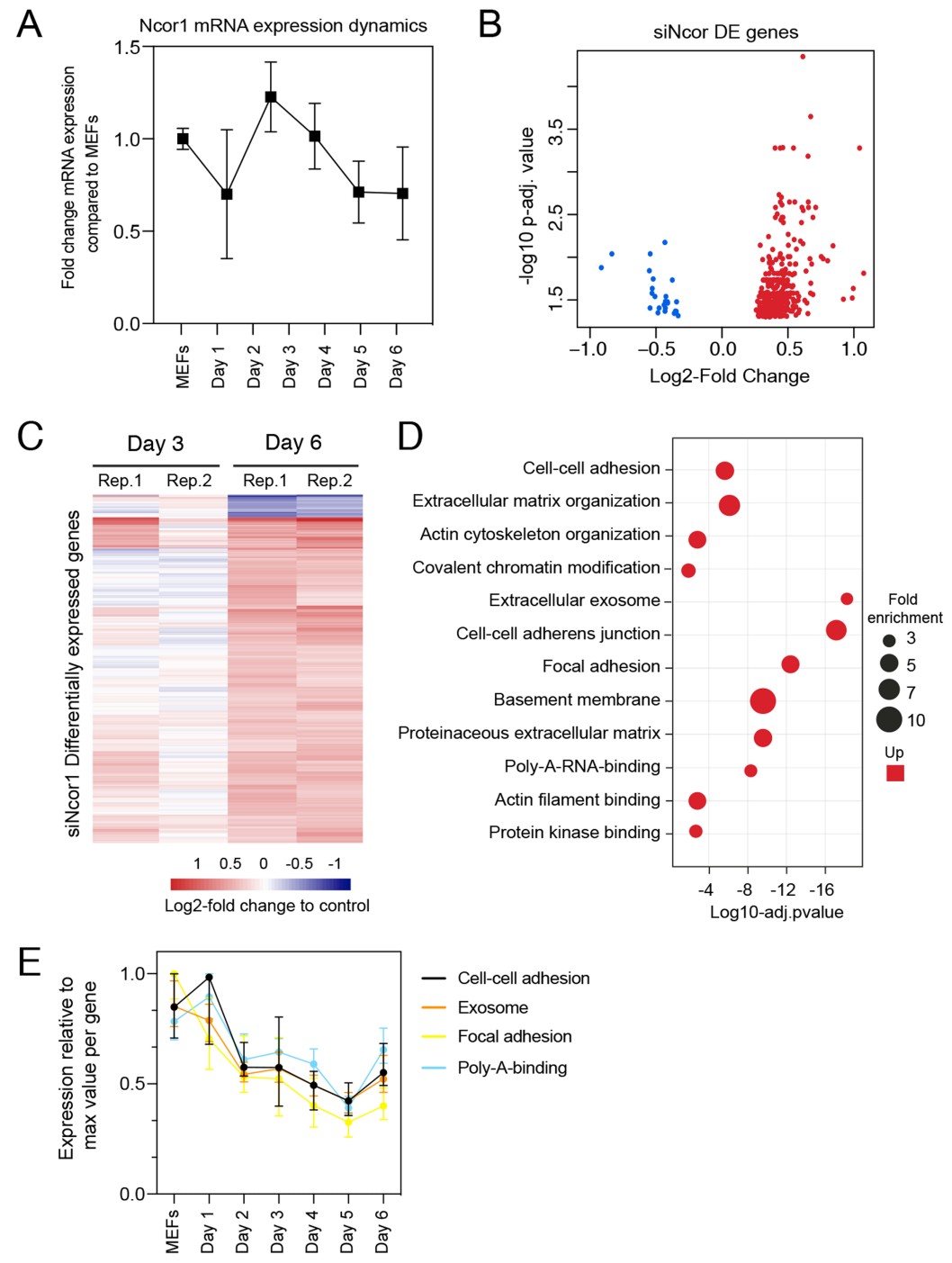

**Figure 2 High content screening identifies *Ncor1* as an early reprogramming modulator.** (A) *Ncor1* mRNA expression dynamics during reprogramming time measured by RT-qPCR. Data are represented as fold change compared to MEFs (day 0). Data points represent the mean ± SD from two replicates from independent samples. (B) Volcano plot for differentially expressed genes in *Ncor1* knockdown at reprogramming day 6, after RNA-sequencing. Blue data points represent the down-regulated genes and red datapoints represent up-regulated genes. (C) Heatmap of differentially expressed genes from *siNcor1* at reprogramming day 3 and day 6. The two lanes labelled as Rep.1 and Rep.2 indicate two biological replicates. Color scale represents the log2-fold change compared to non-targeting control. (D) Gene Ontology (GO) classification from differential genes in C represented as a bubble plot. The *x*-axis of the

**Figure 2** (continued)
plot is the −log10 of the adjusted *p*-value, and some of the most significant terms are represented in red (upregulated). Size of the bubbles represents the fold-enrichment per category. (E) Graphical representation of reprogramming RNA-seq temporal dynamics from some of the upregulated GO categories in (D), shown in different colours. Each datapoint represents the median and the lines the 95% CI. Per GO category, medians were calculated based on the normalised expression value (log2-counts per million reads) of each gene, relative to the gene's maximum value (maximum set to 1).

in reprogramming observed with *siNcor1*, are brought about by moderate changes in expression of hundreds of transcripts (mean absolute $\log_2$-fold change of 0.43). The majority of the genes (94%) were up-regulated, as expected for a transcriptional corepressor, although the deregulation of some of these genes may be an indirect consequence of the reduction of NCOR1 rather than a reduced NCOR1-mediated repression on that particular gene.

To gain insight into which processes were affected in *Ncor1*-depleted cells, we performed Gene Ontology classification (GO) for upregulated genes because the downregulated were very few (Fig. 2D). This analysis showed that upregulated genes were associated with cell–cell adhesion, poly-A-RNA-binding, exosome and cytoskeleton (Fig. 2D). Using our reprogramming RNA-seq time course dataset (*Peñalosa-Ruiz et al., 2019*), we observed that genes associated with these terms were indeed downregulated in the course of early reprogramming (Fig. 2E). To note, the cell adhesion terms featured laminins, collagens and other genes related to basement membrane and extracellular matrix. These data suggest that reduction of *Ncor1* prevents downregulation of genes related to fibroblast identity during MEF to iPSC reprogramming.

## NCOR1 and OCT4 co-regulate two distinct MEF-expressed groups of genes

To gain insight into the functional relationship between OCT4 and NCOR1, we examined RNA-seq profiles at reprogramming day 6 for both knockdowns. To verify *Ncor1* and *Oct4* mRNA-targeting by the corresponding siRNAs, we plotted the RNA-seq normalised values for each of the knockdowns, compared to the controls at day 3 (Fig. S2). *Oct4* knockdown did not show a high efficiency of knockdown at day 3 in these samples, however from time course experiments we have observed that *Oct4* mRNA is targeted and efficiently silenced at day 2, but rapidly recovers within a couple of days (Fig. S2). This is due to its high overexpression driven by the tet-OSKM cassette (*Sommer et al., 2009*). Even with such transient early knockdown we observe a strong phenotype (Fig. 1) and many deregulated genes (Fig. 2A), highlighting the important contribution of OCT4 in OSKM reprogramming.

When we compared genes deregulated in each knockdown we found that 43% of the genes deregulated in *siNcor1* cells were also deregulated in *siOct4* cells (3.4 fold enrichment compared to random overlap, hypergeometric *p*-value $8.9 \times 10^{-54}$; Fig. 3A). This corresponds to only ~5% of genes differentially expressed upon *Oct4* knockdown. After hierarchical clustering of genes deregulated by both *siNcor1* and *siOct4* based on

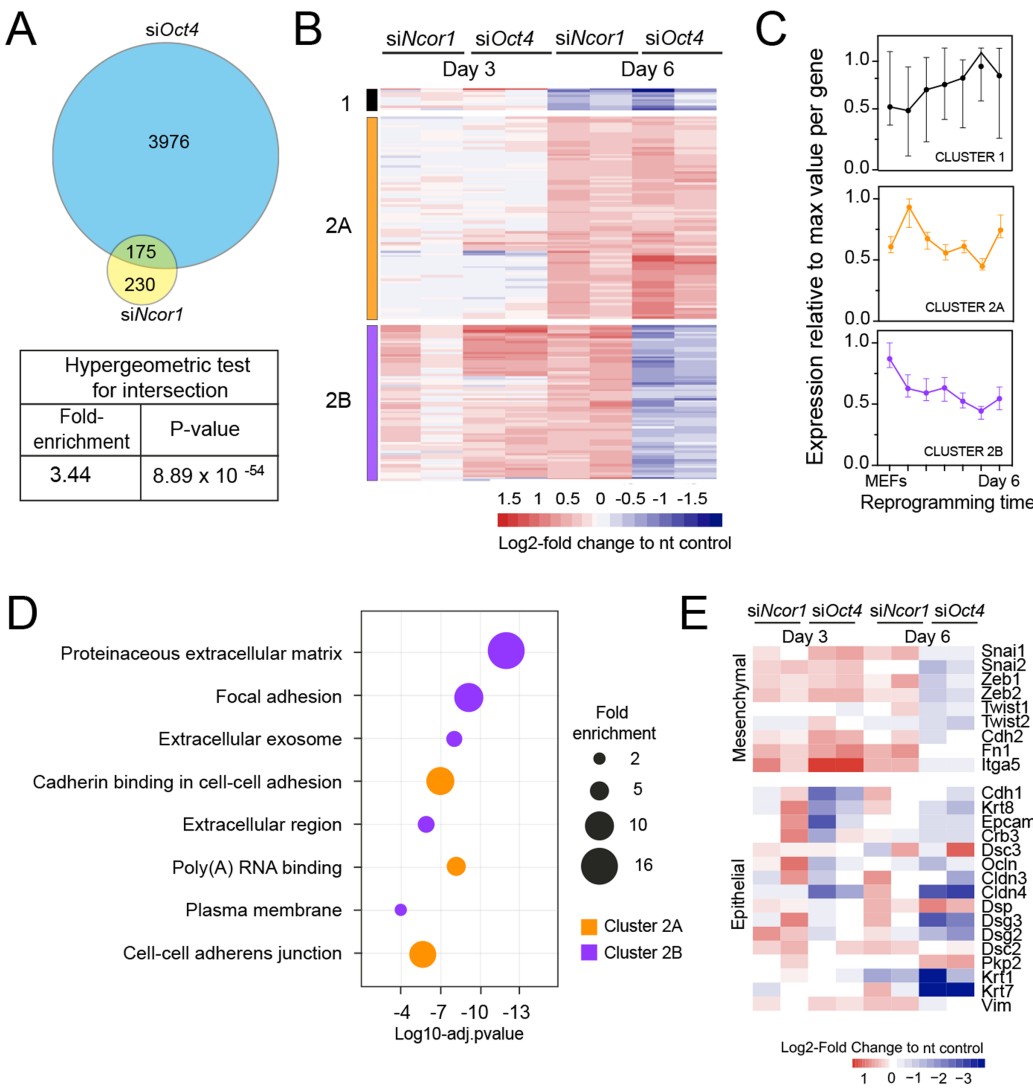

**Figure 3 RNA-seq analysis shows that Ncor1 and Oct4 knockdowns show similar gene expression patterns.** (A) Venn–Euler diagram showing that Ncor1 knockdown shares about half of deregulated genes with *Oct4* knockdown. The table underneath shows values for the hypergeometric test to show that the overlap of *Ncor1* and *Oct4* deregulated genes is significant. (B) Heatmap of shared differential genes between *siNcor1* and *siOct4* from intersection in (A), at reprogramming day 3 and day 6 for both siRNAs. Conditions spanning two lanes correspond to two biological replicates of the experiment. Color scale represents the log2-fold change relative to non-targeting (nt) control. Clusters are tagged in three different colours, cluster 1 in black, cluster 2A in orange and cluster 2B in magenta. (C) Reprogramming RNA-seq temporal dynamics from clusters 1–3 shown in different colours, same colour code as (B). Each datapoint represents the median and the lines the 95% CI. Per cluster, medians were calculated based on the normalised expression value (log2-counts per million reads) of each gene, relative to the gene´s maximum value = 1. (D) Genes shown in clusters 1 and 2 from (C) were subjected to Gene Ontology (GO) classification. The bubble plot indicates some of the most significant classes. The size of the bubble represents the fold-enrichment and the −log10 of the adjusted *p*-value is shown on the *x*-axis. (E) Heatmap representing expression values as log2-fold change ratio to control of mesenchymal and epithelial markers in *Oct4* and *Ncor1* knockdowns at reprogramming day 3 and day 6. The paired columns represent independent RNA-seq samples.

RNA-seq, we identified three clusters (cluster 1, 2A, 2B; Fig. 3B). Cluster 1 genes were relatively unaffected at day 3, but at day 6 they were downregulated for both knockdowns. For cluster 2A we found that genes were also unaffected at day 3, but showed a similar upregulated pattern in both *Oct4* and *Ncor1* siRNAs at day 6 (Fig. 3B). Cluster 2B contained genes with high expression in MEFs, most of which failed to be downregulated at day 3, whereas by day 6 they predominantly showed opposite expression changes in the knockdowns (*siNcor1* vs *siOct4*) (Fig. 3B). Our RNA-seq time course dataset (*Peñalosa-Ruiz et al., 2019*) showed that genes in cluster 1 mostly remain the same in time, whereas clusters 2A and 2B tend to go down in time, but with different dynamics (Fig. 3C).

To shed some light on the difference between clusters 2A and 2B, we performed GO classification for genes of each cluster (Fig. 3D). NCOR1-OCT4 co-regulated genes (cluster 2A) showed an association with RNA metabolism and cell adhesion (Fig. 3D). The term "poly(A)-RNA binding" contained several components for pre-mRNA splicing. Genes increased by *siNcor1* but not *siOct4* (cluster 2B), were associated with functions in extracellular matrix organisation and other cell adhesion features. We wondered whether the "cell adhesion" terms were related to the mesenchymal-to-epithelial transition (MET) observed in early reprogramming (*Li et al., 2010*; *Samavarchi-Tehrani et al., 2010*). We found that most mesenchymal genes exhibit a cluster 2B-type pattern, with increased expression at day 3 in both knockdowns and opposite effects of the two knockdowns at day 6. Epithelial genes on the other hand show more variable patterns (Fig. 3E). In *siOct4* cells, the downregulation of mesenchymal gene expression is delayed. By day 6, it not only catches up, this decrease in expression is more severe in *siOct4* cells relatively to control cells, suggesting that OCT4 moderates this decrease during normal reprogramming. In *siNcor1* cells on the other hand, mesenchymal gene expression is maintained at higher levels at both time points, although some epithelial genes are also expressed at higher levels (Fig. 3E).

These results identify two distinct groups of somatic genes, the expression of which is normally reduced during reprogramming. *Ncor1* depletion prevents downregulation of both groups, whereas *Oct4* depletion prevents the downregulation of one group and causes a stronger decrease in expression in the other group of genes.

## NCOR1 and OCT4 target promoters associated with cellular and metabolic processes in MEFs and early reprogramming cells

To gain insight into the functional relationship between NCOR1 and OCT4, we analysed published ChIP-seq and RNA-seq data (*Chronis et al., 2017*; *Zhuang et al., 2018*). First, we asked whether the up-regulated genes that we identified in *siNcor1* and *siOct4* cells, were either MEF genes or genes dynamically changing during reprogramming. Consistent with our earlier analysis (Fig. 3C), these independent data confirmed that upregulated genes in *siNcor1* cells were mostly MEF-expressed and very early reprogramming (48 h) genes, but were mostly inactive in ES cells (Fig. 4A). In the heatmap several genes related to fibroblast identity are labelled (lamins, collagens, fibronectin, signalling genes; Fig. 4A). In contrast, genes upregulated in *siOct4* cells showed a variety of expression patterns during reprogramming (Fig. 4B).

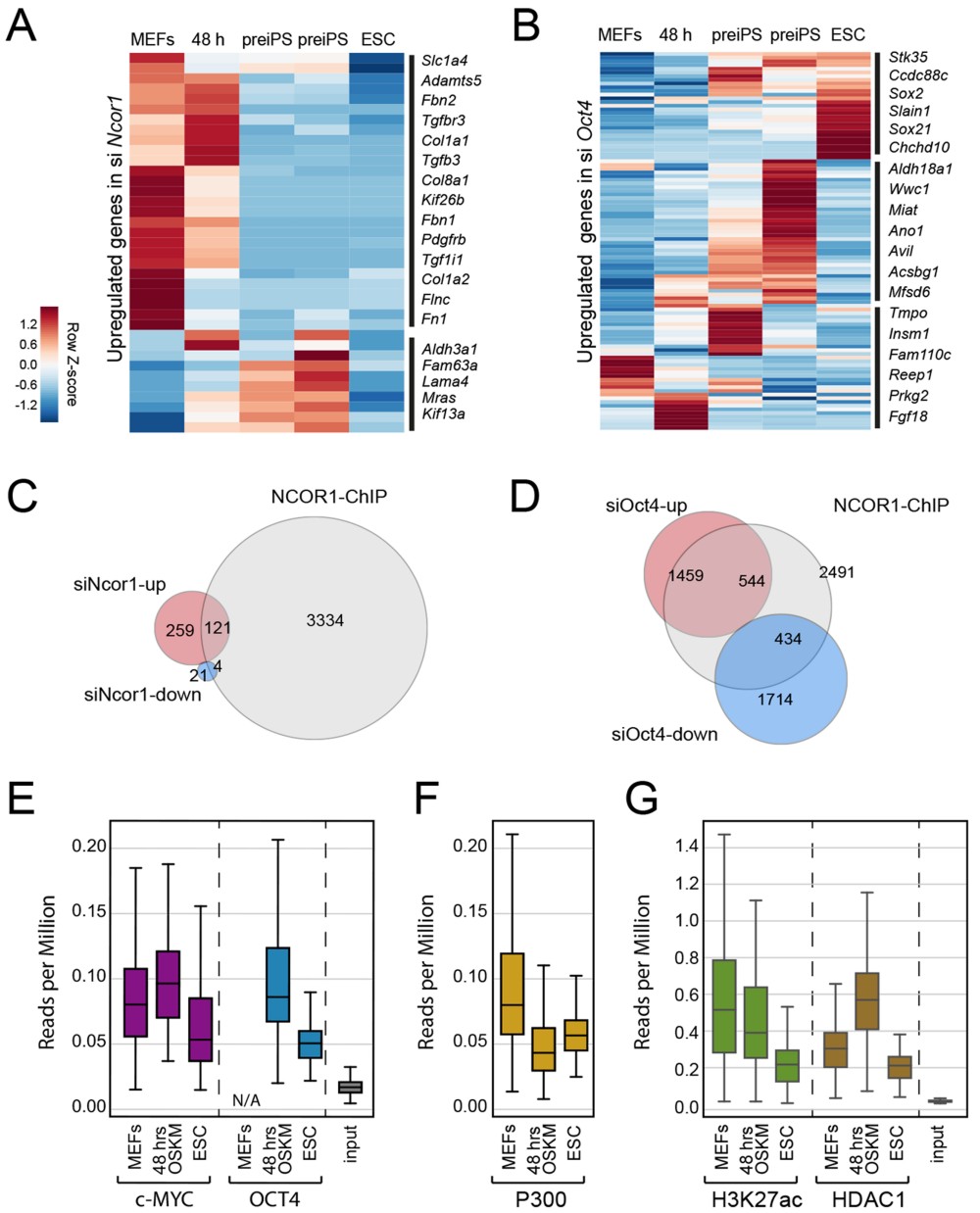

**Figure 4 NCOR1 and OCT4 target promoters in MEFs and early reprogramming cells.** (A) Expression of *siNcor1* up-regulated genes in MEFs, cells after 48 h of reprogramming, pre-iPS intermediates and ESCs. Heatmap representation of differentially expressed genes during reprogramming (*Zhuang et al., 2018*) that are also upregulated by *siNcor1* in our study. The heatmap scale represents row-z scores of normalised expression values. (B) Similar to (A) for siOct4 upregulated genes in our study. Scales in both heatmaps are row Z-scores of normalised gene expression values. (C) Venn–Euler diagram showing NCOR1 ChIP-seq binding sites in reprogramming (*Chronis et al., 2017*), overlapping with up-regulated or downregulated genes in *siNcor1* cells. (D) Venn diagram showing NCOR1 ChIP-seq peaks in reprogramming overlapping with up-regulated or downregulated genes in *siOct4*. (E–G) Boxplots comparing ChIP-seq data for transcriptional regulators MYC and OCT4 (E), P300 (F), H3K27ac and HDAC1 (G). The different colours represent different ChIP antibodies and the lanes are cellular states (e.g. MEFs, ESC, 48 h of OSKM induced reprogramming). The genomic regions in the comparison are genes upregulated in *siNcor1*, overlapping with NCOR1 ChIP binding (intersection of 121 genes in (C)). The scale of the boxplot is given in reads per million.

**Table 1 Overlap of siNcor1 or siOct4 up/down regulated genes with NCOR1 ChIP-seq peaks.** The overlapped was quantified as fold-enrichment and the statistical significance was calculated with a hypergeometric test.

| Overlap comparison | Fold-enrichment | p-Value |
|---|---|---|
| siNcor1 up vs NCOR1 peaks | 3.2 | $5.9 \times 10^{-33}$ |
| siNcor1 down vs NCOR1 peaks | 1.6 | 0.096 |
| siOct4 up vs NCOR1 peaks | 2.7 | $5.3 \times 10^{-115}$ |
| siOct4 down vs NCOR1 peaks | 2.0 | $6.8 \times 10^{-50}$ |

Then, we asked which fraction of differentially expressed genes in *Ncor1*- or *Oct4*-knockdowns were bound by NCOR1 protein. We overlapped NCOR1 ChIP binding sites in reprogramming cells with *siNcor1* and *siOct4* differential genes (Figs. 4C and 4D) and calculated the fold enrichment. We found that NCOR1 binds to a significant fraction of deregulated genes in *Ncor1* and *Oct4* knockdowns (Figs. 4C and 4D; Table 1). It should be noted that the number of genes identified may be affected by the use of siRNA knockdown rather than genetic knockouts and the different time-points and different reprogramming conditions.

To gain insight on the function of transcripts affected by *Ncor1*-depletion in reprogramming, we exclusively selected NCOR1-bound genes that also showed increased expression after *Ncor1* siRNA knock-down (see intersection of 121 genes in Fig. 4C). We used publicly available ChIP-seq datasets to ask whether OCT4 and c-MYC binding was dynamic at these genes during reprogramming (Figs. 4E–4G; Fig. S3). Quantifying the ChIP-seq data (reads per million) showed that c-MYC and OCT4 signal at these 121 genes is higher at 48 h after the onset of reprogramming than it is in mouse embryonic fibroblasts (MEFs) and Embryonic Stem Cells (ESC) (Fig. 4E, MEF data for OCT4 not available (N/A)). This indicates that during early stages of reprogramming OCT4 and c-MYC show increased binding to NCOR1 targets affected by *Ncor1* knockdown. Interestingly, this binding activity is not sustained through the pluripotent state (ESC), hinting at a transient role for NCOR1 in reprogramming.

To investigate this process further, we analysed several chromatin features for the genes in question. In general, these genes contain the active gene marks H3K4me3 (Fig. S3A), indicating some transcriptional activity. Notably, we observed a gradual decrease in signal of the H3K27ac active modification when comparing MEFs, 48 h of OSKM-induced reprogramming and ESCs (Fig. 4G). Moreover, this decrease in H3K27ac levels was concurrent with reduced occupancy of the P300 histone acetyltransferase on these NCOR1 regulated genes (Fig. 4F middle panel). Conversely, binding of histone deacetylase HDAC1 is highest at 48 h after OSKM induction, compared to either MEFs or ESCs (Fig. 4G). These data support the notion that NCOR1 regulated genes are transcriptionally downregulated during the early phases of reprogramming. Additionally, the underlying mechanism may include NCOR1-mediated recruitment of HDAC proteins, including HDAC1. Possibly, reprogramming factors OCT4 and c-MYC could be assisting NCOR1 binding to such genomic targets. However, more experimental evidence is needed to test this.

To explore the function of NCOR1 targets also bound by OCT4 and c-MYC, we performed a Gene Ontology classification of genes analysed in Fig. 4D (Fig. S3). This analysis confirmed once more the terms we observed before (Fig. 2D; Fig. 3D), related to cell-cell adhesion, extracellular exosome, extracellular matrix, cytoskeleton, focal adhesion and poly(A) RNA binding, among others (Fig. S3). As before, the poly(A) RNA binding term included several genes related to translation and pre-mRNA splicing.

## DISCUSSION

In this study we found that depletion of *Ncor1* results in disrupted colony morphology in early iPSC. Transcriptome analyses showed that NCOR1 has a role in suppression of fibroblast identity and signalling modulation (Figs. 2 and 3). Our findings extend an earlier report on the role of NCOR1 in later phases of reprogramming (*Zhuang et al., 2018*). NCOR1 function could be mediated by c-MYC, which occupies many MEF regulatory regions (*Chronis et al., 2017*).

Using multi-dimensional readouts, we found strong phenotypic similarities between *Ncor1* and *Oct4* depletion. Such phenotypic similarities often reflect functional interactions (*Mulder et al., 2012*; *Peñalosa-Ruiz et al., 2019*; *Tanis et al., 2018*). Our analyses indicate that NCOR1 and OCT4 co-bind a group of promoters that are also bound by c-MYC in early reprogramming cell populations. Genes associated with these promoters are expressed in MEFs and during the first 48 h of reprogramming and are mostly transcriptionally silent in ESCs (Fig. 4). Further genomic analyses indicated that these regions are enriched for H3K27 acetylation in MEFs, which dramatically drops in the first days of reprogramming, suggesting a decrease in transcriptional activity. In support of this, the binding of acetyltransferase P300 also decreases from MEFs to 48 h OSKM induction, whereas recruitment of histone deacetylase HDAC1 increases. Since these genes are upregulated upon *Ncor1* knockdown, we could speculate that the repressor NCOR1 is recruited to such promoters by OCT4/c-MYC quite early in reprogramming and transcriptional downregulation occurs via histone deacetylation. Further experiments are needed to elucidate the mechanism. One limitation of these analyses is that the NCOR1 ChIP data corresponds to reprogramming day 9. This time frame does not match those of the rest of the analysed ChIP-seq data. A comparison with matching time points would be ideal for a better interpretation.

There was a surprisingly moderate effect on the transcriptome associated with si*Ncor1*. Because of the increased proliferation of reprogramming cells (*Ruiz et al., 2011*), *Ncor1* knockdown at day 3 was 45%. Perhaps the effects on the transcriptome would have been stronger if we had transfected siRNAs at day 3 instead of day 0, or alternatively with a complete loss of function model. In addition, *Ncor1* and *Ncor2* may have partially redundant functions (*Perissi et al., 2010*), which may have contributed to the relatively mild effect on the transcriptome. Nevertheless, we observe a strong phenotype in colony formation, and the transcriptome data suggest that *Ncor1* attenuates signalling, which might be crucial to weaken fibroblast identity. Consistent with this, NCOR1/NCOR2 are known to be connected to various signalling pathways (*Perissi et al., 2010*).

According to gene ontology analyses, other genes targeted by NCOR1 are related to essential cellular functions, such as cell–cell adhesion, vesicle-mediated transport and RNA splicing. In agreement, complexes and proteins associated with these functions have been shown to be downregulated during intermediate stages of reprogramming (*Hansson et al., 2012*). Moreover some of these processes (e.g. cell adhesion, vesicle-mediated transport) have been shown to be barriers for reprogramming (*Buckley et al., 2012*; *Qin et al., 2014*). One apparent discrepancy in these analyses might be that the active promoter mark H3K4me3 seems abundant in MEFs, 48 h reprogramming cells and ESCs. However, one possible explanation would be that target genes are downregulated in intermediate stages but then required again in late stages, as has been shown for some of the processes involved (cell adhesion, spliceosome) (*Hansson et al., 2012*). Moreover, some of these genes are essential for the cellular functions and continue to be expressed in ESCs, albeit at lower levels.

## CONCLUSIONS

Our high-content data showed that siRNA-mediated depletion of *Ncor1* disrupts iPSC colony-formation capacities and decreases expression of early pluripotency proteins. These phenotypic patterns were very similar to those of *Oct4* knockdown. Therefore, we hypothesised that *Ncor1* was required for early reprogramming and that there might be a functional connexion between *Ncor1* and *Oct4*.

The transcriptome of *Ncor1* knockdown showed a moderate upregulation of fibroblast identity genes. These phenotypic and molecular patterns were very similar to those of *Oct4* knockdown. In agreement, we confirmed that the proteins NCOR1 and OCT4 bind to a common set of regulatory regions. Supporting ChIP-seq data suggested that such regions were transcriptionally active in MEFs (presence of P300 and H3K27ac, low HDAC1) and after 2 days of OSKM induction, they lost enrichment of P300 and H3K27ac, but gained HDAC1. Altogether these data led us to conclude that NCOR1 is involved in the downregulation of fibroblast genes during the early stages of reprogramming. For a subset of these genes, NCOR1 may be assisted by OCT4.

Recent work has shown that NCOR1/NCOR2 interact with all OSKM factors, but especially with c-MYC (*Zhuang et al., 2018*). During late reprogramming stages, the corepressors NCOR1/NCOR2 interact with c-MYC to silence the pluripotency network, posing a barrier for reprogramming (*Zhuang et al., 2018*). Our study uncovers an early facilitating function of NCOR1 in reprogramming, which may suppress cellular and metabolic functions that sustain fibroblast identity, possibly in cooperation with OCT4 and c-MYC. This would be consistent with a role of NCOR1 in overall cellular homeostasis and metabolic regulation (*Fan & Evans, 2015*; *Mottis, Mouchiroud & Auwerx, 2013*). It also highlights the emerging role and cooperation of reprogramming factors in transcriptional repression (*Chronis et al., 2017*).

Future work will focus on potential functional interactions of NCOR1 with other chromatin factors, for instance with PHF1 or APEX1 (Fig. 1B), and their physical interactions in early and late stages of reprogramming.

## ACKNOWLEDGEMENTS

This work was supported by the Microscopic Imaging Center (MIC) from the Radboud University, Nijmegen.

### Funding

The authors received no external funding for this work.

### Competing Interests

The authors declare that they have no competing interests.

### Author Contributions

- Georgina Peñalosa-Ruiz conceived and designed the experiments, performed the experiments, analysed the data, prepared figures and/or tables, authored or reviewed drafts of the paper, and approved the final draft.
- Klaas W. Mulder conceived and designed the experiments, prepared figures and/or tables, authored or reviewed drafts of the paper, and approved the final draft.
- Gert Jan C. Veenstra conceived and designed the experiments, analysed the data, prepared figures and/or tables, authored or reviewed drafts of the paper, and approved the final draft.

### DNA Deposition

The following information was supplied regarding the deposition of DNA sequences:

Data is available at NCBI-GEO: GSE139376.

### Data Availability

Data is available at NCBI-GEO with accession numbers: GSE139376 and GSE118680.

### Supplemental Information

Supplemental information for this article can be found online at http://dx.doi.org/10.7717/peerj.8952#supplemental-information.

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
