# Peer review of "The corepressor NCOR1 and OCT4 facilitate early reprogramming by suppressing fibroblast gene expression"

_PeerJ, doi:10.7717/peerj.8952_

## Round 0.1 · original submission · Minor Revisions

I apologize for the unexpected delay but hope that the reviewers' detailed comments have made the wait worthwhile. I look forward to receiving your revision.

Reviewer 1 ·

Basic reporting

General: This manuscript reports that siRNA-mediated knockdown of the transcriptional corepressor NCOR1 or of the pluripotency transcription factor OCT4 results in similar changes in colony phenotypes in mouse embryo fibroblasts (MEFs) and partially overlapping changes in the transcriptome. The authors also report that NCOR1 and OCT4 co-occupy a shared subset of promoter sites and that in early MEF reprogramming down-regulation of one group of somatic genes requires both NCOR1 and OCT4 whereas another group requires NCOR1 but not OCT4. Later in MEF reprograming a separate cluster of genes are also upregulated by siNcor1 but are downregulated by siOct4, suggesting the expression of the latter genes is normally facilitated by OCT4. The authors conclude that “NCOR1, assisted by OCT4 and C-MYC, facilitates transcriptional inactivation of genes with high expression in MEFs, which need to be suppressed to bypass an early reprogramming block.”

Overall this is a fairly straightforward, concise manuscript that reports several interesting observations relevant to how pluripotency in MEFs is regulated. The analysis is carefully performed, the results are generally convincing, and most of the conclusions are well supported by the experiments provided. However a few suggestions are proposed to improve the presentation and to test one of the authors' hypotheses (please see specific comments).

The manuscript is clearly written and the introduction, background, and citations are appropriate and well-constructed. The figures are relevant and are sufficient quality, adequately labeled and described (with just a few exceptions; please see specific comments).

Experimental design

The research question, the role of NCOR1 vs OCT4 in control of pluripotency, is well defined and is appropriate for PeerJ; its relevance to previously unanswered questions is clearly explained. The technical and ethical standards appear high and the methods are sufficiently detailed. The submission website provides the various usual assurances but I was unable to find a statement in the text stating these. Will this be reconciled on publication?

Checks: The submission website indicates that files were deposited to NCBI-GEO (which I confirmed) but I was unable to find a statement in the text stating this. Will this be reconciled on publication? MEFs, not a cell line, were employed. The use of English is very good. My inspection of the data reveals no obvious manipulation but my ability to test this is very limited.

Validity of the findings

The experiments are generally well designed scientifically, adequately controlled, and the results are fairly straightforward. The authors’ conclusions are clearly stated and generally well founded on their experimental results, although one of the authors' hypotheses would be strengthened by one or two additional modest experiments; several other conclusions would benefit from being discussed at greater length (please see specific comments).

Specific Comments: 1. The authors provide several lines of evidence that NCOR1 and OCT4 coregulate an shared subpanel of genes, including an apparent overlap between NCOR1 and OCT4 promotor binding sites by ChIP analysis. The results presented are generally supportive of this argument but there are some incompletely resolved questions in this regard:

a. Figure 4D presents ChIP occupancy data for different proteins/chromatin modifications at different times post-induction; unfortunately the time point presented for the NCOR1 ChIP does not match those of the other ChIPs. The power of these comparisons would be significantly strengthened if the authors had access to, or could newly generate, data from a matched time course NCOR1 CHIP. Alternatively (recognizing that the NCOR1 ChIP is not their own data and that de novo ChIP experiments would require significant effort and time) a discussion of the limitations of the current data may be adequate instead. The authors should also discuss why, if only a minority of NCoR-occupied sites exhibit up or down regulation in response to siOct4 (Figure 4C) all of the NCoR1 occupied/regulated sites presented in Figure 4D also exhibit co-occupancy by OCT4: are the genes presented in this panel already filtered to present only those upregulated by both siNcor1 AND siOct4?
b. The authors state that “NCOR1 directly binds to a significant fraction of deregulated genes in Ncor1 and Oct 4 knockouts” (lines 328-329). The most likely mechanism for this observation and for the reported overlap between NCOR1 and OCT4 occupancy sites in Figure 4D, would be if the repressor NCOR1 “is recruited to such promoters by OCT4/c-MYC" (lines 364-365). Have the authors tested this hypothesis in more detail such as by (i) a “reChip” analysis using anti-OCT4 antibodies followed by anti-NCOR1 antibodies, and/or by (ii) a coimmunoprecipitation of OCT4 and NCOR1 from MEFs after DNase I treatment or an in vitro GST-pulldown protocol using the two proteins? These would also help confirm/define the term “direct” in line 328.
2. It is curious that only ~4% of the sites of Ncor1 binding identified by ChIP in Figure 4C are up regulated by siNcor1, and more than 2/3's of the genes upregulated by NCor1 siRNA are not associated with an Ncor1 binding site in the ChIP. These observations do not invalidate the authors’ conclusions but the authors definitely should discuss possible mechanisms that might explain these results and their implications.

3. What was the efficiency of the siNcor1 knockdowns in the experiments presented in this manuscript?

4. The following are related to presentation rather than the validity of the science itself.

a. The image in Figure 1D is too dark and too small to easily evaluate by the reader. Can better panels be provided?

b. The labelling of the heat maps in Figures 2 and 3 confused me. Each descriptor at top (e.g. "siNcor1" or "siOct4" in Figure 2) spans two lanes of heat map but the difference between those two lanes is not labeled or explained in the captions. Does each lane represent a separate set of genes within the same overall category (e.g. cluster 2A, Day 6, siNcor1)? Or do the two adjacent lanes represent the same genes under different conditions not specified by labeling?

c. I could not find legends for the Supplemental figures.

d. A few statements in the text should be reworded to be more precise:
(i) Line 271-272, “approximately half of the genes deregulated in siNcor1 cells were also deregulated in siOct4 cells…” The actual percent is 43%; why not just use that value rather than round up to "approximately half"? It should also be explicitly stated here (and in lines 400-404 of the Conclusions) that, reciprocally, less than 1/20 of the genes responding to Oct4 siRNA also respond to siNcor1. The asymmetry in these NCOR1 vs OCT4 overlaps is important in considering the mechanisms that may be operative and the conclusions that can be made; it should be noted prominently in the Results and specifically addressed in the Discussion.
The following are relatively minor points:
(ii) Line 78, “NCOR1/NCOR2 are essential enzymatic co-factors of histone deacetylases.” The construction of this statement is slightly ambiguous and may lead some readers to incorrectly assume that NCOR1/NCOR2 are themselves enzymes, rather than non-enzymatic cofactors for the enzyme HDAC3.
(iii) Line 328-329, “NCOR1 directly binds to a significant fraction of deregulated genes in Ncor1 and Oct 4 knockouts…” Technically the siRNAs generate "knockdowns" not "knockouts." Use of the term “directly” is discussed in specific comment 1b, above.

Reviewer 2 ·

Basic reporting

Well written and presented, adequate references in general, clear hypothesis and well-supported data.

Experimental design

Good in general.

Validity of the findings

Appropriate and timely.

Additional comments

In the manuscript by Penalosa-Ruiz et al., the authors continue their previous work based on a high content siRNA screen on the reprogramming of somatic cells to induced pluripotent stem cells. In this screen, they discovered a link between NCOR1 and OCT4, which they now study in detail and go on to validate using a range of genomic tools. The author’s screen was set up in such a way that it is more likely to get hits during the early phase of reprogramming, which is a particularly interesting phase as the mechanism behind the suppression of somatic genes still remains somewhat fuzzy and there is much disagreement. Hence, the observation that OCT4 can recruit NCOR1 to suppress somatic and fibroblastic genes adds a useful dimension to the action of the reprogramming factors in reprogramming, and contributes to the wider discussion on the role of epigenetic control in reprogramming and cell fate conversions. The results are consistent with a previous manuscript by Zhuang et al. in Nature Cell Biology (2018) showing that NCOR/SMRT contribute to suppress the somatic cell program during reprogramming, but the present work extends those observations significantly and provides new data too.

Overall, on a technical level I found the study to be well performed, and technically sound. Statistical analysis is appropriate. I have a few relatively minor points the authors should address.

1. Figure 4D. As NCOR1 modulates H3K27ac, it would be useful to add the H3K27ac data from Chronis et al. here to see if these loci are indeed marked as active in MEFs. 
2. The authors don’t seem to show the level of knockdown of NCOR1 by RT-qPCR and in the RNA-sequencing data. In a related point, does the knockdown of OCT4 affect NCOR1? Also, what are the expression levels of NCOR2 in the various knockdowns?
3. Since NCOR1 plays a dual role in reprogramming (inhibits both the somatic cell program and pluripotency genes) it would be useful if the authors could explain that siRNA and shRNA knockdowns may turn the balance towards increased or decreased reprogramming efficiency (full reactivation of the pluripotency network at late time points, which is not measured here) differently depending on the strength of the knockdown.

Minor stylistic points:

1. Line 52-54. Whilst a very attractive model, it is by no means proved that chromatin state is the dominant controller for cell type stability. Should be rewritten in a more speculative tone.
2. Line 67. It is not clear that OCT4 binds to open chromatin first. It could bind both closed and open simultaneously or the timing for one process or the other be a matter of the time points assessed or be due to differences in chromatin state in mouse and human reprogramming, and/or be caused by the type of medium used for the reprogramming.
3. Please correct issues with terminology. For example, NCOR2 is written as NCOR2, SMRT, and SMART depending on the paragraph. Check figures as well. Also, there is iPSC and iPS. Moreover, check gene versus protein terminology.

---

## Round 0.2 · accepted · Accept

The care taken by the reviewers was reflected by the care taken by the authors in addressing their points and concerns both in their discussion and in their additional integration of data with previous literature. It was a pleasure to accompany this article, fostering the cordial and productive exchanges that have led to a new functional contribution to the stem cell literature.